# Ranking parameters driving siring success during sperm competition in the North African houbara bustard

Gabriele Sorci [1✉], Hiba Abi Hussein[2], Gwènaëlle Levêque[3], Michel Saint Jalme[4], Frédéric Lacroix[2], Yves Hingrat[2] & Loïc Lesobre [2]

Sperm competition is a powerful force driving the evolution of ejaculate and sperm traits. However, the outcome of sperm competition depends on many traits that extend beyond ejaculate quality. Here, we study male North African houbara bustards (*Chlamydotis undulata undulata*) competing for egg fertilization, after artificial insemination, with the aim to rank the importance of 14 parameters as drivers of siring success. Using a machine learning approach, we show that traits independent of male quality (i.e., insemination order, delay between insemination and egg laying) are the most important predictors of siring success. Traits describing intrinsic male quality (i.e., number of sperm in the ejaculate, mass motility index) are also positively associated with siring success, but their contribution to explaining the outcome of sperm competition is much lower than for insemination order. Overall, this analysis shows that males mating at the last position in the mating sequence have the best chance to win the competition for egg fertilization. This raises the question of the importance of female behavior as determinant of mating order.

[1] Biogéosciences, UMR 6282 CNRS, Université de Bourgogne, Dijon, France. [2] Reneco International Wildlife Consultants LLC, Abu Dhabi, United Arab Emirates. [3] Emirates Center for Wildlife Propagation, Missour, Morocco. [4] Centre d'Ecologie et des Sciences de la Conservation, CESCO, Museum National d'Histoire Naturelle, CNRS, Ménagerie le zoo du Jardin des Plantes, Sorbonne Université, Paris, France. ✉email: gabriele.sorci@u-bourgogne.fr

When females mate with multiple males during the same reproductive bout, sperm from different individuals compete to fertilize the eggs[1,2]. By affecting reproductive success, sperm competition exerts selection on many characteristics of the ejaculate, and several lines of evidence show that when males compete for egg fertilization, they appear to invest more in traits that improve their likelihood to outcompete the rivals[3,4]. In its simpler form, sperm competition is supposed to favor males that transfer the largest number of sperm during copulation (the raffle principle)[5]. However, it rapidly became clear that the outcome of the competition among ejaculates was not only a matter of numbers[6], and several sperm phenotypic traits and ejaculate attributes were identified as predictors of male siring success[7–11]. Even considering both sperm number and quality does not provide a complete picture of the complex interactions that take place among competing sperm within the female reproductive tract[12–14]. Although females generally provide a selective environment for sperm[15], some environmental features (e.g., ovarian fluid) might be more favorable to one male's sperm than others, biasing the reproductive success towards specific males[16]. Differences in the similarity at specific genetic loci (MHC) between the female and the competing males can also result in skewed siring success in favor to the most dissimilar mates[17]. Other variables also play a major role to determine who will win the contest. In internal fertilizers, females do not mate simultaneously with different males and matings always occur in order. Depending on how sperm are stored in the female reproductive tract, or the rate of sperm loss, male order is often identified as a good predictor of siring success under controlled laboratory conditions[12], with last males in the sequence having the best siring success (i.e., the last male sperm precedence)[18]. The strength of last male sperm precedence under natural mating conditions is, however, less clear, given that all other traits potentially affecting fertilization success also vary among males across the mating sequence[18]. Nevertheless, if the last male sperm precedence operates under natural conditions, it might have the potential to weaken the selection acting on ejaculate and sperm traits, especially so in species where females have a direct control over mating order (because they can choose the male with whom they mate) or an indirect control (because they can reject the sperm of undesired males after mating)[19]. In addition to mating order, other traits likely unrelated to male ejaculate quality might shape siring success. For instance, if mating occurs too distantly from egg laying, sperm can suffer from an age-dependent decline in the fertilization capacity due, for instance, to oxidative damage[20]. Finally, if males with preferred phenotypic attributes mate more often than less preferred ones, their ejaculates might get depleted, finally reducing their chance to fertilize the eggs[21].

This short overview illustrates how focusing on ejaculate attributes while neglecting other factors likely to affect male siring success might provide a biased assessment of the strength of selection acting on sperm traits during sperm competition. However, we lack a comprehensive analysis of the relative importance of the different traits that can determine male siring success (but see[22]). Here, we took advantage of a large dataset that has been collected over many years on the siring success of captive male North African houbara bustards (*Chlamydotis undulata undulata*) and, using a machine learning approach, we ranked the importance of 14 parameters that might a priori be linked to the outcome of the among-male competition for egg fertilization, following artificial insemination of females. Our analysis showed that traits unrelated to male quality, such as male order in the mating sequence, are by far the most important drivers of siring success during sperm competition.

## Results

We used a mixed boosted regression trees (BRT) model to rank the importance of several potential predictors of male fertilization success. BRT is a machine learning technique that combines regression trees and boosting to improve the model predictive power (see methods for details). We first checked whether the model had a good enough predictive power, depending on how the dataset was split into training and testing. We used a nested cross-validation (CV) strategy to simultaneously tune the hyperparameters, train, and evaluate the model performance (see methods for details). The variability of the prediction performance among the 5 folds of the cross validation was low, showing consistency between folds and insensitivity to the splitting of the data (Table 1). The average predictive performance of the BRT model was relatively good with an average accuracy of 71% (SD = 2.2) implying that the model was able to correctly identify the male who fertilized the egg in 71% of the cases. The other metrics assessing model performance were also consistent with a good agreement between predictions and observations [MCC = 0.39 (SD = 0.05); ROC-AUC = 0.77 (SD = 0.02)].

We identified 15 potential predictors of male fertilization success, following artificial insemination of females (Table 2). However, due to the redundancy between two parameters (see methods for details), 14 predictors were finally included in the BRT model. The 14 predictors encompass traits describing male and ejaculate quality (e.g., number of sperm in the ejaculate), and parameters unrelated to male quality (e.g., male order in the insemination sequence). The importance of each parameter as predictor of male fertilization success was assessed based on SHapley Additive exPlanations (SHAP), a game theoretic approach allowing to interpret the output of machine learning models. The model included 2226 insemination events (with 879 males having contributed semen to these inseminations) and 901 eggs laid by 599 females.

The SHAP summary plot of the model shows that male position in the insemination sequence was the predictor with the highest contribution to predict siring success (mean absolute SHAP value = 0.674) (Fig. 1). Males whose sperm was used last in the insemination sequence were associated with positive SHAP values and consequently were predicted to have higher siring success compared to males in the other positions of the insemination sequence (Fig. 1). The partial dependence plot showed that the probability to fertilize the egg drastically decreased between males who were last in the insemination sequence and the preceding position, while for males who were in position >2 the relationship flattened (Fig. 2a; note the inversed order, 1 referring to the last male in the sequence).

The second most important predictor of siring success was the delay between the day of egg laying and the day when the insemination occurred (mean absolute SHAP value = 0.341) (Fig. 1). Inseminations occurring between 4 and 12 days prior to egg laying were those with the highest probability to fertilize the egg (Fig. 2b).

Albeit of lower importance compared to insemination order and delay between egg laying and insemination, the two variables describing the quality of the ejaculate (number of sperm in the ejaculate and mass motility index) ranked high (3rd and 4th) among the different predictors (mean absolute SHAP values = 0.091 and 0.089, respectively) (Fig. 1). The probability to fertilize the egg increased with increasing number of sperm in the ejaculate and mass motility index, and the relationship saturated for high values of both variables (Fig. 2c and Supplementary Fig. 1).

The investment into a secondary sexual trait (percent of days displaying prior to the day the ejaculate was collected), supposedly involved in pre-copulatory sexual selection, ranked fifth among the predictors (mean absolute SHAP value = 0.045), with

**Table 1 Grid search hyperparameter optimization results and fivefold cross-validation model performance evaluation.**

| | LearningRate | Max. Depth | Min. Data in Leaf | Number of trees | CV performance | | |
| --- | --- | --- | --- | --- | --- | --- | --- |
| | | | | | Accuracy | MCC | ROC – AUC |
| Fold 1 | 0.01 | 5 | 5 | 1337 | 0.73 | 0.43 | 0.78 |
| Fold 2 | 0.01 | 5 | 100 | 1378 | 0.72 | 0.41 | 0.77 |
| Fold 3 | 0.01 | 5 | 100 | 1230 | 0.69 | 0.33 | 0.77 |
| Fold 4 | 0.01 | 3 | 50 | 1240 | 0.74 | 0.45 | 0.80 |
| Fold 5 | 0.01 | 3 | 10 | 1568 | 0.69 | 0.35 | 0.75 |
| All data model | 0.01 | 3 | 50 | 1812 | | | |
| | | | | Average | 0.71 | 0.39 | 0.77 |
| Grid space | {0.01, 0.005} | {3, 5, 7} | {5, 10, 20, 50, 100} | Max = 2000 ESC = 10 | Standard Deviation | 0.02 | 0.05 | 0.02 |

*Max Maximum number of trees, ESC Early Stopping Criterion, CV Cross Validation, MCC Matthews Correlation Coefficient*

Fifty random hyperparameter combinations were tested during the grid search and the number of iterations (trees) was tuned simultaneously using a maximum threshold of 2000 trees and an early stopping criterion set to 10 which will stop the training process if the model performance does not improve after 10 consecutive iterations.

a negative relationship with siring success (Fig. 1). Males with the highest siring success were those with the lowest investment into sexual display, and siring success declined as investment into sexual display increased (Fig. 2d).

The number of inseminated sperm, the delay between the day the focal insemination was performed and the previous insemination, and the delay since male's previous ejaculate were all in agreement with the predicted relationship with siring success (positive, positive, negative; Supplementary Fig. 1), but the contribution of each of these variables to the model prediction was modest (mean absolute SHAP values ranging between 0.044 and 0.029, Fig. 1).

The date of the insemination ranked at an intermediate position (mean absolute SHAP value = 0.036). Siring success tended to increase for inseminations performed later in the season, but the relationship flattened as the season progressed (Supplementary Fig. 1).

The number of breeding events in captivity recorded along the pedigree (a possible proxy of adaptation to captivity) ranked relatively low in the overall predictor list (tenth position) (mean absolute SHAP value = 0.027). However, the relationship was negative, potentially suggesting a cost of adaptation to captivity in terms of impaired siring success during sperm competition (Supplementary Fig. 1).

Finally, the inseminated volume, the total number of ejaculates collected prior to the one used to inseminate the female, and male and female age appeared to have a nil contribution to siring success (mean absolute SHAP values ranging between 0.017 and 0.000) (Fig. 1, Supplementary Fig. 1).

## Discussion

We showed that last male sperm precedence and sperm aging in the female reproductive tract are the two most important predictors of siring success during sperm competition in the North African houbara bustard. Although lower in the ranking of predictors, traits describing (i) ejaculate quality (i.e., number of sperm in the ejaculate and mass motility index); (ii) investment into a secondary sexual trait; (iii) and possibly genetic consequences of captive propagation were also associated with differential probabilities to fertilize the egg. Finally, male age and traits potentially describing sperm depletion did not seem to play a role as drivers of siring success, within the range of values explored.

We assessed the predictive performance of our model using different metrics and all of them consistently provided evidence suggesting a good agreement between observed and model predicted values. For instance, an average ROC-AUC of 0.77 reveals a fairly good model fit and predictive ability, especially if we consider that other traits not included in our model can still contribute to explain among-male variation in siring success (e.g., compatibility for specific genetic loci between males and females).

The ranking based on SHAP values suggested that last male sperm precedence is the most important determinant of siring success. Actually, among the whole sample of eggs included in our dataset, males in the last position of the insemination sequence fertilized 63% of them. The mechanisms accounting for the competitive advantage of males who mate at the last position of the mating sequence are manifold. For instance, in certain species, specific structures of the male genitalia allow to remove sperm deposited in the female reproductive tract during previous matings (e.g.,[23]). However, since the houbara bustard lacks an intromittent organ, this seems unlikely. Alternatively, if sperm stratify in the female sperm storage tubules, or are displaced during successive matings, sperm from the last copulating male might have a better access to the eggs[12]. Finally, the

**Table 2 List of predictors of siring success under competitive inseminations included in the BRT model.**

| Predictor | Mean | SD | Range | Description | Reason underlying the predicted association between the predictor and siring success | Expected relationship |
|---|---|---|---|---|---|---|
| Male insemination order | 1.87 | 0.9 | 1–6 | Inversed position in the insemination sequence (1 referring to the last male in the sequence) | Last male sperm precedence | |
| Delay between insemination and egg laying | 11 | 7 | 1–30 | Delay in days between the laying date of the egg and the day the ejaculate was collected and used for the insemination | Sperm aging in the female reproductive tract | |
| Delay since male's previous ejaculate | 3.79 | 4.22 | 0–57 | Delay in days between the day the ejaculate was collected and used for the insemination and the day the previous ejaculate was collected | Sperm aging in the male reproductive tract and/or sperm depletion | |
| Delay with previous insemination | 3.2 | 4.2 | 0–23 | Delay in days between the day the ejaculate was collected and used for the insemination and the day of the previous insemination | The longer the delay the higher the likelihood for the last male to compete against aged sperm | |
| Number of ejaculates collected | 17 | 11 | 0–54 | Total number of ejaculates collected during the same year preceding the one used for the insemination | Sperm depletion | |

**Table 2 (continued)**

| Predictor | Mean | SD | Range | Description | Reason underlying the predicted association between the predictor and siring success | Expected relationship |
|---|---|---|---|---|---|---|
| Male age | 4.24 | 2.3 | 1–21 | Age in years of the male | Organismal aging | |
| Number of male breeding events in captivity | 6.4 | 5 | 0–33 | Total number of breeding events in captivity, recorded along the pedigree, necessary to produce the male | Adaptation to captivity | |
| Number of male generations in captivity | 2.44 | 0.83 | 0–4.92 | Male generation in captivity computed as the average generation of its parents plus one[54] | Adaptation to captivity | |
| Number of sperm in the ejaculate | 43 | 28 | 5–248 | Number of sperm in millions in the collected ejaculate | Ejaculate quality | |
| Mass motility index | 3.8 | 0.61 | 1–5 | Mass motility index of the collected ejaculate, ranging from 0 to 5[50] | Ejaculate quality | |

**Table 2 (continued)**

| Predictor | Mean | SD | Range | Description | Reason underlying the predicted association between the predictor and siring success | Expected relationship |
|---|---|---|---|---|---|---|
| Number of inseminated sperm | 19 | 10 | 2–99 | Number of sperm in millions used for the insemination | Insemination quality | |
| Volume inseminated | 99 | 38 | 13–250 | Total volume in microlitres used for the insemination | Insemination quality | |
| Percentage of days displaying | 33.47 | 19.11 | 0–86.06 | Percentage of days the male was observed displaying between January 1st and the day the ejaculate was collected and used for the insemination | Secondary sexual trait | |
| Day of insemination | 93 | 21 | 33–165 | Date of the focal insemination in days (1 = 1st January) | Seasonal effects | |
| Female age | 4.65 | 2.94 | 1–22 | Age in years of the female | Male–female compatibility | |

For each predictor, we report its description, the mean, standard deviation (SD) and range. We also describe the reason underlying the inclusion of each parameter as a predictor of siring success and the expected relationship between the predictor and siring success. Note that the expected relationship only refers to the sign of the effect and not to the shape of the relationship, which might not necessarily be linear. For predictors related to adaptation to captivity, investment into secondary sexual traits, and date, we did not have a unique expected relationship with siring success. Female age cannot have a direct effect on fertilization success (thus the expected relationship is a flat line), because we only included fertilized eggs in the dataset and any female trait would be an invariant with respect to fertilization success. However, including female age might improve the predictive power of male traits due to male–female compatibility effects. N = 901 eggs, 2226 inseminations, 879 males and 599 females.

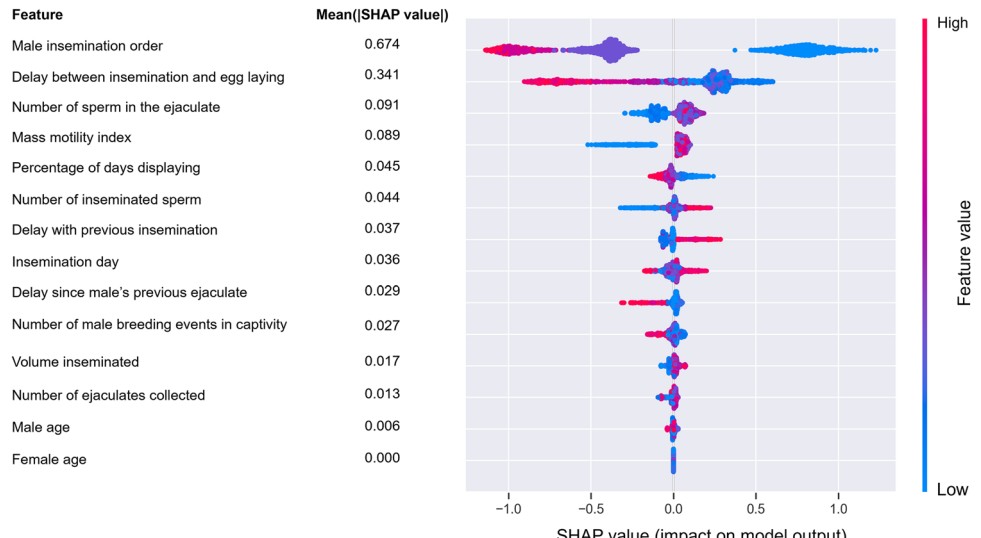

**Fig. 1 SHAP summary plot of the 14 predictors of siring success included in the BRT model.** The variables on the vertical axis are ranked according to the mean of the absolute SHAP values, indicating their relative global importance as predictors of siring success (higher mean absolute SHAP values corresponding to a larger contribution to the siring success prediction). Dots represent the SHAP values (positive values being indicative of higher siring success, lower values being indicative of higher siring failure) for each insemination ($N = 2226$) as a function of the value of the predictor (the redder the color the higher the value of the predictor). Please note that for the insemination order, 1 refers to the last male in the sequence.

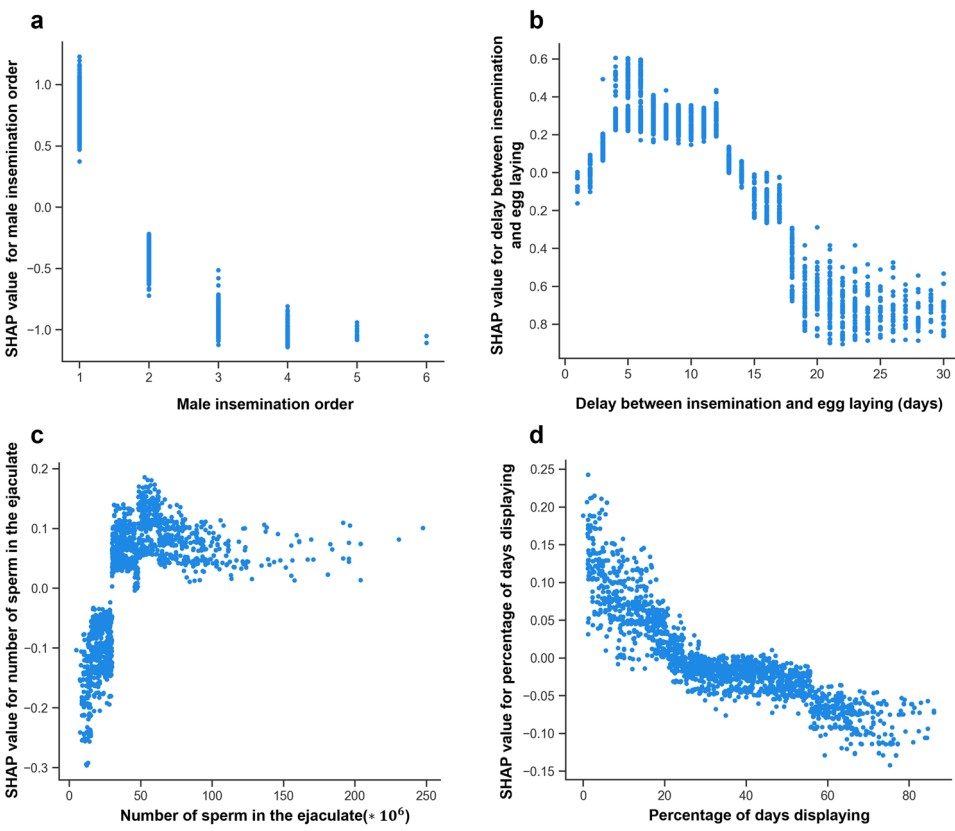

**Fig. 2 SHAP partial dependence plot of the main predictors of siring success. a** Male insemination order (1 refers to the last male in the sequence); (**b**) delay between the day of egg laying and day of insemination; (**c**) number of sperm in the ejaculate; (**d**) percentage of days displaying prior to the day the ejaculate was collected. Each dot corresponds to one insemination ($N = 2226$).

rate of sperm loss with time might explain why last copulating males have a competitive advantage[24]. This is even more likely when consecutive matings occur several days apart from each other.

The rate at which sperm are lost can be constant with time or can accelerate while sperm age[25]. Sperm aging has been identified as an important determinant of fertilization success because sperm are particularly susceptible to oxidative damage due to their high

metabolic activity and poor antioxidant defences[20,25,26]. Sperm age during their storage in the male reproductive tract (until they are released in the ejaculate) and during their storage in the female reproductive tract (until they are used to fertilize the egg). Previous work has shown that prolonged storage both in the male and female reproductive tract is associated with reduced fertilization and impaired offspring quality[27–29]. We assessed the importance of sperm aging using three proxies. The delay between the day of egg laying and the day of insemination refers to the aging in the female sperm storage tubules; the delay between the day of collection of the ejaculate and the day of collection of the previous ejaculate refers to the aging during storage in the epididymis; the delay between the day of the insemination and the day of the previous insemination refers to the difference in age between competing sperm. Overall, our results support the hypothesis that these different aspects of sperm aging do play a role during competition for egg fertilization. In agreement with the predicted sign of the relationship, we found that the longer the storage in the female and in the male reproductive tract, the lower the probability to fertilize the egg. Similarly, when the delay between two consecutive inseminations was long, the competitive advantage of the last insemination was rather substantial, possibly due to the impaired fertilization ability of sperm that had been stored for longer periods, to passive sperm loss, and the combination of the two.

The ecological relevance of both last male sperm precedence and sperm aging as drivers of siring success in natural populations is difficult to assess, and has been discussed in previous work[18]. The reason is that experimental manipulation of mating order is usually done while keeping constant other individual traits whose variability is potentially associated to siring success. Results obtained in the lab might therefore poorly reflect the natural situation where males with different phenotypic attributes can mate over the whole range of mating orders. Similarly, assessing sperm aging in the wild is not straightforward due to the necessity to control for the duration of sperm storage in male and/or female reproductive tract[27]. While we acknowledge that our approach is still far away from the natural situation, the artificial insemination of females with a wide range of ejaculate attributes and other parameter values provides an ideal opportunity to get as close as possible to the interactions that are likely to occur in nature. Interestingly, our results on the importance of last male precedence are broadly consistent with those recently reported by[22] for natural populations of red junglefowl (*Gallus gallus*).

Albeit of lesser importance, intrinsic ejaculate attributes were also associated with the outcome of sperm competition. In particular, males producing large ejaculates with highly motile sperm had a clear advantage when competing for egg fertilization. These findings corroborate previous work conducted in this and in other species and are consistent with the classical paradigm of sperm competition where investment into ejaculate quality (e.g., number of sperm, sperm motility, viability, etc.) is associated with enhanced siring success[7–9,11]. Surprisingly, the number of sperm in the ejaculate ranked higher than the actual number of sperm inseminated, although the shape of the relationship between fertilization success and each of these two predictors was strikingly similar. The reason why the number of sperm in ejaculate had a higher predictive power might be due to a positive covariation between this trait and other sperm and/or ejaculate attributes that we did not measure and that might affect fertilization success. Overall, the finding that ejaculate attributes are still good predictors of siring success even when other traits (e.g., male mating order) are explicitly taken into account in the model, strongly suggests that selection on these traits might operate in the wild.

Broadly speaking the sources of variance in male reproductive success can be decomposed into two sequential stages. First, males have to acquire a mate; second, once males have successfully mated, their sperm have to fertilize the egg. Failure at any of these stages implies a nil reproductive success. Maximizing access to mate and egg fertilization relies on different phenotypic attributes. For instance, pre-copulatory sexual selection is often supposed to be driven by the expression of secondary sexual traits involved in male-male competition and/or female choice, while post-copulatory sexual selection often involves investment into ejaculate attributes[2,30]. Whether males can simultaneously optimize investment into traits promoting pre- and post-copulatory sexual selection has been extensively discussed, and several studies have reported either positive or negative phenotypic (and genetic) correlations between traits involved in the two stages of sexual selection[31]. We were able to include a trait referring to the investment into pre-copulatory traits in our modeling approach. Male houbara bustards perform a complex sexual display during the breeding season on dedicated sites[32]. This sexual display involves feather ornaments, behavioral traits, and vocalizations (booming)[33,34]. Although our study was conducted under captive conditions, males still perform their display in the aviaries and therefore we could assess a (crude) proxy of male investment into pre-copulatory traits as the percentage of days males were observed displaying between the beginning of the season and the day when the ejaculate was collected. This proxy of investment into pre-copulatory sexual selection ranked at the 5th position among the full list of predictors, just after the two traits referring to ejaculate quality. The SHAP values showed that the siring success was the highest for those males with the lowest percentage of days with display preceding the ejaculate collection. This result might indicate a physiological trade-off between investment into traits promoting access to mate and traits ensuring maximum fertilization success, as reported in other species[33].

An important concern related to conservation breeding programs is the risk for individuals to adapt to the captive conditions over generations. This might impair their capacity to succeed under the harsher environmental conditions they experience once released into the wild[35]. Simultaneously, involuntary selection for improved fecundity and fertility might occur, and especially so at the start of the program when the flock still counts a relatively low number of individuals[36]. We found some weak evidence suggesting that males with longer history of captive breeding tended to have a reduced siring success compared to males with a more recent history of captivity. Along with adaptation to captivity, captive propagation can also lead to loss of genetic diversity through genetic drift, inbreeding or relaxation of natural selection[37,38]. In the houbara bustard captive breeding program, a strict genetic management is implemented and neither loss of genetic diversity nor increased inbreeding has been reported[39]. However, in the long run, the genetic management might also produce a relaxation of selection acting on males, potentially contributing to explain the reduced siring success of males with the longest history of captivity. That said, we would like to emphasize that the overall contribution of this effect of captivity was rather small, suggesting that the risk of jeopardized reintroduction success due to impaired sperm competitive ability is minimal.

Apart from female age, male age was the predictor with the least (virtually nil) contributing importance for siring success. This result might appear surprising for several reasons. First, there is extensive evidence that has been accumulated over the last years showing that reproductive senescence is a pervasive phenomenon in nature[40]. Male reproductive senescence is usually associated either to impaired access to mates or impaired capacity to fertilize the egg (with carry over effects of paternal age extending to the progeny as well)[41,42]. Male reproductive senescence has already been reported in previous work conducted in

the North African houbara bustard[43–45], including experiments where sperm from young and old males were forced to compete for egg fertilization[46]. Based on these previous findings, we had strong a priori reasons to expect male age to be high in the ranking. However, it turns out that due to the inclusion criteria used to select the data, the distribution of male age was very skewed towards young males. Of the 879 males included in the dataset, 90% of them were ≤ 7 years. Knowing that prime age for reproductive performance is around 4–6 years in the species[43], the number of senescing individuals in the dataset was clearly too small to be able to see any age effect.

Our study was conducted under captive conditions, which allowed us to assess several parameters that would have been impossible to investigate under natural conditions. At the same time, we would like to stress that the environmental conditions encountered under captivity and the specific design used to breed the birds (artificial insemination) do not allow us to conclude that these results would necessarily apply for free ranging birds. Precluding females to express a pre-copulatory choice (which is inherent to artificial inseminations) might also affect how sperm are used. Similarly, the expression of male sexual display might be affected by the amount of available resources (which certainly vary between captive and natural conditions), or inter-individual differences in the way birds cope with captivity. With this respect, it should be noted that all individuals forming the captive flock are born in captivity and human imprinted which reduces the stress due to captivity.

Our work is based on a retrospective analysis of data that have been routinely collected, independently from the specific aim of this study. Therefore, the values of some of the predictors included in the model might span over ranges larger than those observed in the wild. Unfortunately, very little information is available on the mating behavior of the houbara bustard in the wild. For instance, we do not know how many times females mate before laying, how long is the delay between mating and egg laying, how often males mate, etc. The development of new technologies (remote accelerometer sensors) should help to fill this gap.

We provided here a rather comprehensive assessment of the main factors potentially shaping male siring success during competitive fertilizations. These factors covered a large spectrum of ecological and physiological traits, ranging from the expression of pre-copulatory traits to male age. The finding of a predominant importance of last male sperm precedence in this species raises the question of whether, under natural conditions, the order of mating occurs at random or is under female control. With a few exceptions, birds do not have intromittent organs, therefore males have little opportunities to remove sperm from the female reproductive tract. On the contrary, evidence has been reported suggesting that female behaviors that are expressed both pre- and post-mating, might contribute to determine the order at which sperm of different males compete[22]. This might contribute to partially shifting the force of selection from male ejaculate traits to female behaviors.

## Methods

**Model species**. The North African houbara bustard (*Chlamydotis undulata undulata*) lives in arid lands across North Africa. The species is sexually dimorphic in both size and plumage (males are bigger and harbor longer ornamental feathers). Its breeding season extends from January to June. Males express a conspicuous sexual display on dedicated sites ("exploded" leks)[32], and females mate with several males, as shown by the occurrence of multiple paternities within broods[47]. Sperm of different males, therefore, regularly compete for the fertilization of eggs. However, detailed information on how many times females and males mate, the frequency of mating, how long before laying females mate, etc. is missing for natural populations. In the wild, females typically lay one to three egg(s), and can produce a replacement clutch if the first one is lost[48]. Only females provide parental care, and chicks remain with their mother for 6 to 10 weeks[49].

**Captive breeding program**. The decline of natural populations of the North African houbara bustard, mainly due to over-hunting and habitat degradation, prompted the creation, in 1996, of the Emirates Center for Wildlife Propagation (ECWP) based in eastern Morocco, with the aim of restoring populations throughout the species range in North Africa. Birds of the captive flock are individually housed in sheltered outdoor pens (2 × 2 m) with food and water ad libitum.

**Ejaculate collections and artificial inseminations**. Birds used in this study are part of the ECWP. The program entirely relies on controlled artificial inseminations[50] and the management of the captive flock is based on strict rules aiming at equalizing representation of founders, minimizing inbreeding, and maintaining genetic variation[51].

The semen of males constituting the captive flock is routinely collected using a dummy female. A dummy female is presented to males and the ejaculate collected in a Petri dish held under the male cloaca. This should ensure that the quantity of semen collected closely approaches the amount ejaculated during a natural copulation (at least compared to other semen collecting techniques, such as massage). Ejaculates are transferred into an Eppendorf tube (and the total volume of the ejaculate measured) and brought immediately to an adjacent laboratory where semen quality is assessed at room temperature. First, sperm motility is scored from 0 to 5 under a light microscope using the following scale:(0) total lack of movement, (1) few motile sperm without forward movement, (2) less than 50% of sperm showing moderate activity, (3) above 50% motile sperm showing forward movement, (4) above 80% motile sperm showing fast forward movement, (5) almost all sperm showing fast forward movement with waves and whirlwinds[50]. Then, the number of sperm in the ejaculate is assessed using a spectrophotometer and a specific calibration for the species[52].

Based on their laying history, females are regularly checked by zootechnicians throughout the breeding season and when deemed ready to lay, they are inseminated through the slow deliver of sperm inside the vagina using a positive displacement pipette[50]. For each insemination performed, the number of sperm and the volume inseminated is recorded.

Before the first egg of the clutch is laid, two sequential inseminations are performed with a 48 h interval (but occasionally females can lay after the first insemination); a third, fourth (and so on) insemination can be added if required (i.e., if the first egg of the clutch takes longer to come than expected)[53]. Therefore, depending on when the egg is laid, females can be inseminated once, twice, or more. Such successive inseminations can involve sperm collected from the same male or from different males [with each insemination involving sperm from a single male (e.g., no mix of sperm was used)][53]. When females are successively inseminated with sperm from different males, sperm competition can occur[53].

Each egg is collected on the day it is laid and brought to the incubation facility where it is weighted and incubated. At hatching, chicks are weighed and then raised following specific protocols, depending on whether birds are deemed to be released in the field or kept in the captive flock as future breeders.

**Paternity assignment**. When females are sequentially inseminated with semen from different males, sperm competition can occur. In this case, siring success is assigned based on the genotyping of 9 microsatellite loci designed for the houbara bustard[47]. A small volume of blood (~100 μL) is taken from all birds (candidate fathers, mother, and offspring) and stored in 2 mL Eppendorf tubes containing 1.5 mL of absolute ethanol. DNA extraction and genotyping were carried out by GENOSCREEN (Lille, France), the detailed procedure being described elsewhere[47]. Paternities are assigned using CERVUS 3.0.

**Courtship display**. During the breeding season, captive males exhibit a complex courtship display combining conspicuous visual and acoustic components similar to those observed in the wild. This energy-demanding behavior of males is monitored for all males by the staff of the ECWP, and the absence or presence of display is recorded. These observations therefore provide a crude proxy of male investment into pre-copulatory traits.

**Adaptation to captivity**. Animals can get adapted to the specific conditions experienced in captivity. In addition, involuntary selection can favor individuals with the highest fertility/fecundity, especially when the captive flock still counts few individuals. Adaptation to captivity should increase as the number of generations experiencing the captive conditions increases. Given that the pedigree is known for all the birds of the captive flock, for each individual included in the dataset, we computed the number of generations in captivity as the average number of generations of its parents plus one[54]. However, in unbalanced pedigrees (i.e., where all branches to the founders are not of the same length), for the same number of generations, individuals may have been produced following different numbers of breeding events in captivity, which can also contribute to the process of adaptation to captivity. We therefore computed the number of breeding events in captivity that were necessary to produce the males included in the dataset. Supplementary Fig. 2 illustrates the differences between the two parameters within a complete pedigree.

**Criteria for data inclusion**. Using the database with all the records of female inseminations, we retrospectively identified eggs that were laid following

inseminations with sperm from different males, corresponding to cases where sperm competition could have occurred. We used the following rules to decide whether an egg could be included in the dataset: (1) the female was inseminated with sperm from at least two different males in the 30 days that preceded egg laying; (2) each male contributed to a single insemination per female; (3) paternity was assigned based on microsatellite genotyping. These inclusion rules were based on the following rationale. Female houbara bustards can store sperm in dedicated structure and use them well after insemination has occurred[53]. However, the probability of egg fertilization decreases as a function of the time elapsed between insemination and egg laying; therefore, we set a conservative limit to 30 days before egg laying for data inclusion. Sometimes, females are inseminated several times with semen from the same male, which inevitably skew the probability of egg fertilization towards this male; therefore, we only considered siring success when each competing male contributed to a unique insemination. Finally, when multiple males compete for egg fertilization, paternity cannot be assigned with certitude unless the genotype of putative fathers and offspring is known; therefore, we only included eggs for which microsatellite genotyping has been conducted for all competing males and chicks. A total of 901 eggs and 2226 insemination events met the inclusion criteria. These eggs were laid by 599 females while 879 different males contributed semen for the inseminations. The average fertilization success per male (success/attempts) was 38.7% (SD = 40.12%).

**Predictors of male siring success.** For each egg included in the dataset based on the criteria explained above, we had information on several parameters that might a priori contribute to explain among-male variation in siring success and refer to different aspects/stages of the interaction between competing sperm:

(1) Male order in the insemination sequence (last male sperm precedence).
(2) Delay between the day the insemination occurred and the day of egg laying (sperm aging in the female reproductive tract and/or passive sperm loss).
(3) Delay between the day the ejaculate was collected and used for the insemination and the day the previous ejaculate was collected (sperm aging in the male reproductive tract and/or sperm depletion).
(4) Delay between two consecutive inseminations (the possible handicap of aged sperm competing with freshly inseminated sperm)
(5) Number of ejaculates collected prior to the one used for the insemination (since January 1st) (sperm depletion).
(6) Male age (organismal aging).

(7) Number of breeding events in captivity recorded along the pedigree (adaptation to captivity).
(8) Number of generations spent in captivity (adaptation to captivity).
(9) Number of sperm in the ejaculate (intrinsic ejaculate quality).
(10) Mass motility index (intrinsic ejaculate quality).
(11) Number of inseminated sperm (quality of the insemination).
(12) Volume inseminated (quality of the insemination).
(13) Percentage of days the male was observed displaying between January 1st and the day the ejaculate used for the insemination was collected (investment into secondary sexual traits)
(14) Day of insemination (variation occurring across the breeding season).
(15) Female age (male–female compatibility).

Therefore, 15 parameters (but see below for the reasons underlying the exclusion of one of them) were considered as possible predictors of siring success during sperm competition. January 1st was set as the starting day of the breeding season. Note that female age was included to explore the possibility that compatibility between males and females might contribute to explain fertilization success (e.g., through age-matched fertilization success). Given that only fertilized eggs were included in the dataset, female age is an invariant which, therefore, cannot have any direct effect on fertilization success. Also note that each predictor was evaluated separately for each insemination by a particular male (i.e., we never used previous values from that male). Table 2 reports the full list of predictors (with mean and range) and for each of them the expected effect on siring success.

**Statistics and Reproducibility.** Including a large number of variables in the same model can produce several statistical issues. First, it increases the risk of collinearity[55]. The inspection of the correlation matrix between the predictors described above revealed strong Pearson correlations between two pairs of variables: number of male generations in captivity and number of male breeding events in captivity ($r = 0.868$); male order in the insemination sequence and delay between the day of egg laying and the day the insemination occurred ($r = 0.796$) (Fig. 3). Second, investigating potential interactions between variables or possible nonlinear relationships between the predictors and the response variable results in a multidimensional model with high complexity, prone to computational and convergence issues.

To overcome these problems, we used a mixed boosted regression trees (BRT) model[56]. BRT is a machine learning method whose features make it more suitable

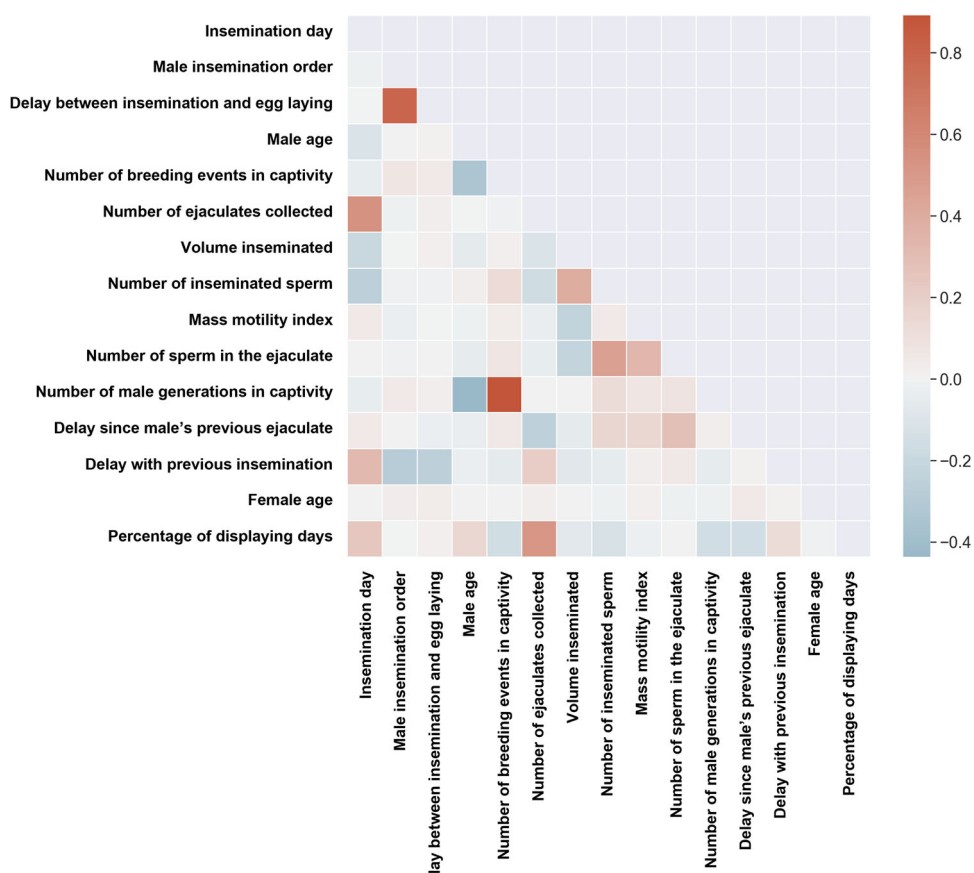

**Fig. 3 Covariation between the predictors of siring success.** Correlation heatmap (squares below the diagonal report the Pearson correlation coefficients) of the 15 predictors of siring success. $N = 879$ males and 599 females.

compared to generalized linear mixed models. BRT is insensitive to outliers, predictor scaling, missing values, and collinearity. Moreover, the contribution to the predicted siring success provided by BRT for each variable entered into the model takes into account the potential interactions among predictors and any nonlinear relationship between predictors and the response variable, ensuring a strong predictive power.

BRT combines classification and regression tree models with boosting optimization techniques. Based on a provided number of predictors, it builds and sequentially combines a high number of weak learners (or decision trees) with each learner attempting to correct for the errors of the previous one[57], the aim being to determine the best combination of features that provides a robust estimator and consequently an accurate classification. Our dataset corresponds to clustered data where the dependency between the different inseminations competing for the fertilization of the same egg and the repeated use of ejaculates from the same male to inseminate different females have to be modeled while building the boosted trees. Therefore, we used the Gaussian Process Booster implemented in GPBoost python algorithm[58]. GPBoost uses LightGBM library[59] for tree learning and gradient descent for the covariance parameter learning. Since our main goal was to rank the predictor importance, we decided, despite the robustness of boosted trees to collinearity, to drop one variable of the pair of the highly correlated variables (correlation coefficient = 0.868), as they seem to provide redundant information. Therefore, the predictor "number of generations in captivity" was discarded and the final number of predictors was 14.

Hyperparameter tuning is a key step for the algorithm to build a robust classifier and to avoid over- or under-fitting. The main hyperparameters are: (i) learning-rate (lr) which is the factor that scales the contribution of each tree to the growing model; (ii) the number of boosting iterations corresponding to the number of trees to be built and controlling for the fit of the tree, (iii) the tree depth and (iv) the minimal number of samples in a leaf that regulate together the complexity of the trees. We used a nested cross-validation (CV) strategy to simultaneously tune the hyperparameters, train, and evaluate the model performance. It consists in two nested loops where the outer loop split the dataset into $k$ equally sized and non-overlapping partitions (here $k = 5$). The data of the males competing for the fertilization of the same egg were included in the same dataset. On the $k − 1$ training sets, fivefold cross validation was performed by the inner loop to tune the model hyperparameters through a random grid search-based algorithm to select the best combination, among 50 random combinations, to be used for model training. Then, a fivefold cross validation was used by the outer loop to train the model and evaluate its performance on the held-out test set. At the end of the outer loop, $k$ models were fitted and evaluated, and the averaged predictive performance metrics were reported such as the prediction accuracy, Matthew's correlation coefficient (MCC)[60] and the area under the receiver operating characteristic curve (ROC-AUC)[61]. The accuracy measures the proportion of correct predictions (True Positive + True Negative) out of the total number of predictions. MCC is a robust metric that measures the correlation between the binary actual data and the model predictions[62]. The ROC-AUC measures the separability degree of a model between the 2 classes.

Finally, the SHapley Additive exPlanations (SHAP) technique[63] was used to interpret how the predictions of the complex black-box model were performed. We chose this method due to its superior performance compared to other interpretation approaches such as Local Interpretable Model-agnostic Explanations (LIME)[63]. The SHAP approach has the advantage of being mathematically very consistent, accurate and able to reveal reliable hidden relationships between predictors[64]. SHAP values are generated based on game theory to describe and rank the importance of each variable based on its contribution to the prediction of the response variable (here the binary variable "fertilization success"). The programming language Python (version 3.10.2)[65] was used to build the mixed boosted trees model and to generate the SHAP plots. The complete code is available in the Supplementary Information.

To check the consistency of the ranking provided by the BRT model, we also ran a generalized linear mixed model with a binomial distribution of errors, and the identities of the male and the egg as crossed intercept random effects. In this model, we only included the five most important predictors, according to the BRT ranking, as fixed effects. The predictors were standardized (mean = 0, SD = 1), which allowed us to rank them according to the parameter estimates. The output of this model was consistent with the ranking of the BRT, with insemination order and delay between insemination and egg laying having the largest effect sizes (Supplementary Table 1).

**Reporting summary**. Further information on research design is available in the Nature Portfolio Reporting Summary linked to this article.

## Data availability
The data underlying all the results presented here are available on DRYAD and can be retrieved from https://doi.org/10.5061/dryad.9w0vt4bkv[66].

## Code availability
The python code use to run the BRT model and compute the SHAP values is available in the Supplementary Information.

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

## Acknowledgements

Funds and samples used in this study were provided by the International Fund for Houbara Conservation (IFHC). We are grateful to His Highness Sheikh Mohamed bin Zayed Al Nahyan, President of the United Arab Emirates and founder of the IFHC, His Highness Sheikh Theyab bin Mohamed Al Nahyan, Chairman of the IFHC, and His Excellency Mohammed Ahmed Al Bowardi, Deputy Chairman, for their support. This study was conducted under the guidance of Reneco International Wildlife Consultants LLC, a consulting company that manages the IFHC's conservation programs, including ECWP. We thank all staff of Reneco who participated in data collection.

## Author contributions

Conceptualization: G.S., L.L., M.S.J., Y.H. Data curation: L.L., H.A.-H. Formal analysis: H.A.-H. Funding acquisition: F.L. Project Administration: G.L., F.L. Supervision: G.S., Y.H. Visualization: G.S., H.A.-H. Writing—Original draft preparation: G.S., H.A.-H. Writing—Review & editing: G.S., H.A.-H., L.L., G.L., M.S.J., F.L., Y.H.

## Competing interests

The authors declare no competing interests

## Ethics approval

The captive breeding runs under the approval of Moroccan authorities: Ministère de l'Agriculture, Développement Rural et des Pêches Maritimes, Direction Provinciale de l'Agriculture de Boulemane and Service Vétérinaire (Nu DPA/48/285/SV) under permit number 01-16/VV; OAC/2007/E; Ac/Ou/Rn.
