## [Peer Review File · Communications Biology]

Reviewers' comments:

Reviewer #1 (Remarks to the Author):

In their paper 'Ranking the importance of factors driving siring success during sperm competition', Sorci and colleagues used a large dataset from a long-term breeding program in houbara bustards for a comprehensive assessment of the factors potentially contributing to competitive fertilization success. They included a wide range of traits reflecting ejaculate quality, ejaculation and insemination history, and male age or sexual display and used a powerful statistical approach to estimate the relative contributions of these different parameters to the paternity of each egg. Male insemination order and female sperm storage duration before fertilization had the strongest effect on paternity, followed by sperm motility and number.

This study makes an important contribution to the field by showing the multifarious nature of competitive fertilization success without the rigorous experimental control of confounding variables (but possibility of statistical control), which potentially captures a more natural scenario than in previous studies.

Some points I would like the authors to consider are:

1. The authors mentioned male-female interactions (or female biases among sperm from different males) in their introduction, which is an important issue but not addressed in the analyses.
 - a. Is there any information on female age, fecundity, reproductive history or other that could be examined along with the male traits?
 - b. Do some females consistently show a stronger bias between males? Are the replicated combinations between males and females that could be compared (e.g., repeatability within combinations, consistent ranking between combinations)?
2. In many species, males have been shown to plastically adjust sperm number or quality to their social context, be it presence of other males or female quality.
 - a. How many different dummy females were used and how do they compare?
 - b. Is there any effect of dummy ID on ejaculate traits that could indicate strategic ejaculate allocation and thus variation between samples?
 - c. If not, do you have any data on natural matings that would allow you to infer the importance of male-female combinations?
 - d. Finally, are males exposed to other males before or during semen collection that could affect the samples?
3. The authors repeatedly refer to 13 or 14 'phenotypic' traits that they measured. However, I would not consider variables like mating order, Julian date or (artificial) insemination quality 'phenotypic traits'. How about 'variables' or 'parameters', or else counting the phenotypic traits and those simply used as control variables?

Some more specific comments (by line):

L26 'of' instead of 'from' after independent

L40 It would be fair to acknowledge Parker (1970) here, too

L43 'contest' sounds a bit odd for sperm competition. The also, due to the generality of this phenomenon, you could also consider citing some broad reviews (e.g., Pizzari and Parker 2009; Parker and Pizzari 2010) instead of a random mix of intraspecific and comparative case studies (which themselves are not necessarily directly comparable).

L48-50 Note that a follow-up paper by the same team even specifically addressed those complex interactions between competing sperm and the female reproductive tract (Lüpold et al. 2020).

L50 I think 'hostile' should be avoided as it is a very loaded term. How about 'selective', 'challenging' or anything along those lines? If you choose the former, it also makes it immediately clear why some males' sperm might be favored over others as mentioned in the rest of this sentence.

L138 Better: "For each egg included..."

L187 'technique' needed'? If yes, plural? Does not quite work as it stands.

L201 'Hyperparameter tuning'

L332 True, but more likely this enhanced advantage could result from the combination of sperm aging and sperm loss.

L388 'released into the wild'

L390 'start'

L397 'in the long run'

Literature cited:

Lüpold, S., J. B. Reil, M. K. Manier, V. Zeender, J. M. Belote, and S. Pitnick. 2020. How female × male and male × male interactions influence competitive fertilization in *Drosophila melanogaster*. *Evol. Lett.* 4:416–429.

Parker, G. A. 1970. Sperm competition and its evolutionary consequences in the insects. *Biol. Rev.* 45:526–567.

Parker, G. A., and T. Pizzari. 2010. Sperm competition and ejaculate economics. *Biol. Rev.* 85:897–934.

Pizzari, T., and G. A. Parker. 2009. Sperm competition and sperm phenotype. Pp. 207–245 in T. R. Birkhead, D. J. Hosken, and S. Pitnick, eds. *Sperm Biology: An Evolutionary Perspective*. Academic Press, San Diego.

Reviewer #2 (Remarks to the Author):

«Ranking the importance of factors driving siring success during sperm competition» uses an impressive dataset on artificially-inseminated houbara bustards, from a well-studied long-term captive breeding program, to examine how various factors relate to fertilization probability after competitive inseminations. The paper is well-written, and it provides a valuable contribution to understanding how different factors related to ejaculate timing and quality, and male quality, affect fertilization success in a bird. As the authors note, this type of data is quite rare for a long-lived vertebrate. They use a machine learning approach, which they explain clearly, to investigate this. I am not qualified to comment on the accuracy/rigor of the machine learning approach.

1. I recommend that the authors present some follow-up traditional statistics and/or figures, perhaps as a supplement, to help ensure readers understand how to interpret the SHAP outputs (and also to be sure that readers are convinced by these outputs!) The authors justify the machine learning approach partly as a way to avoid difficult problems with multicollinearity, but it seems to me that one or two smaller, confirmatory model based on the machine learning results could be useful. For example, a model including either insemination order or insemination-laying delay (as they are highly correlated, Fig 2), as well as sperm motility and/or the sperm number in the ejaculate (which are more weakly correlated), and number of days the male displayed. Some of these variables might be converted to categorical variables based on the machine learning results (e.g., number of sperm in the ejaculate). Of course it is the variance inflation factor, rather than the raw correlation among these variables, that is important; perhaps including all 5 predictors in one model would be possible.

2. In a few places, the paper over-sells its ability to reflect selection on sperm in the wild (e.g., the contrast between lab and natural conditions at L 58-60; the fact that neither the abstract, the introduction, nor the opening paragraph of the discussion states that the study uses artificial insemination or places caveats on the findings related to that; L 334-339). While the authors do acknowledge that their study is a long way from wild conditions (L 341-349), several aspects of the study are over-interpreted with respect to their immediate relevance to wild populations, and this must be improved.

2A. For example, L 355-385 focuses on the proxy for pre-copulatory investment. In the wild, among-male variation in the ability to acquire resources could alter how (and if) investment into pre- and post-copulatory traits are correlated (see, e.g., Kvarnemo & Simmons, 2013), while in the captive population, presumably there is only variation in allocation. It's therefore difficult to directly apply these results to wild animals. Also, L109-113: Please provide evidence that this measure reflects investment into pre-copulatory sexual selection, rather than, for example, variation in how well males tolerate captivity.

2B. Additionally, several variables relating to timing seem likely to be longer than in nature. Specifically, Table 1 indicates that the delay between insemination and egg laying is on average 11 days, which is quite a lot longer than in passerine birds (e.g., Johnsen et al., 2012). The number of days between ejaculations by the male is quite variable (up to 57 days, mean of about 4), and the delay between inseminations is on average 3.2 days, with a maximum of 23. Is there any information about these variables from wild populations of bustards, so that we know whether the values used are biologically relevant in a wild condition? Delay between successive inseminations only seems to have an effect when it has a value greater than about 8 days (Fig S1C), which is also about the point where inseminations become inherently less effective (delay between insemination and egg laying, Fig 3B), which suggests possible collinearity problems.

2C. Finally, one additional element that should be mentioned is the possibility that female interactions with the male himself guide how females use his sperm (mentioned in general, e.g., L 66; see also citation 16 Løvlie et al.).

3. In addition, two variables should be altered and re-analyzed, and the authors should consider including additional measures of intrinsic ejaculate quality, such as morphological normality of the sperm (e.g citation 11, Vuarin et al. 2019).

3A. L 149, table 1: The number of ejaculates collected within the preceding year seems like much too broad a time span for examining sperm depletion. In Bengalese finches (admittedly a distant relative), recovery from sperm depletion occurs within 24 hours (Birkhead, 1991). I hope it is feasible to re-calculate this variable using a shorter time window; otherwise, please do not make conclusions about the role of sperm depletion from this variable (e.g., L 296).

3B. L 378-384, 109-113: Consider using the percent of days, rather than the number, when the male displays (from the beginning of the season), to avoid dependence between these two variables. If the number of days the male displays largely reflects the date when the sperm was sampled, it is difficult to interpret this as a proxy for investment into precopulatory traits.

4. At L 413-414, we learn that the males were relatively young. Are there other ways in which the

dataset is a biased subsample of males/inseminations? For example, in citation 59, the authors describe an experiment aimed at discovering age effects, which might result in the competing ejaculates being from more age-different males than would be expected by chance. Are there other similar factors at play?

5. It strikes me as strange that the number of sperm in the ejaculate, but not the number of sperm inseminated, is a predictor of fertilization success (L259, 274). This suggests that it's not sperm number per se, in the sense of a sperm competition raffle, that is driving the pattern, but perhaps some other ejaculate quality trait that correlates with number of sperm ejaculated? This result is not highlighted in the discussion.

Minor comments

L 25, 163: not all are phenotypic traits (e.g., mating order, date); perhaps "factors" would work instead?

L 26: Similarly, I was confused at first reading about what you meant by independent from male quality (particularly since mating order might correlate with male quality in wild populations, and the abstract does not state that this was a study using artificial insemination). It might read better if you instead re-word and describe all 3 non-intrinsic factors that were important.

General methods section (or perhaps in the supplement): please describe briefly the housing conditions of the birds. Are males isolated from females? What kind of enclosure are they in?

L 90-92: Clarify that the ECWP is in Morocco, to clarify why the animal care approval from Morocco is the relevant approval

L99-100, 132-134: Can you please provide information about how many males' sperm was inseminated for each egg, and perhaps give a range of what percent of "attempted" eggs each individual male succeeded in fertilizing?

L105: "on the day of laying, all eggs" implies that the female lays multiple eggs in one day. Is that correct for this species? If not, perhaps reword to "Each egg is collected on the day it is laid..."

L 134 (or elsewhere): please clarify whether predictor variables (e.g. sperm motility index) were evaluated separately for each insemination by a particular male (not using a previous value from that male)

L 142-162, Table 1: The authors have done a really good job of keeping organized with so many different predictor variables. This could be further improved if they are presented in the same order in the text and in the table. "delay with previous ejaculate" might be more clear as "Delay since male's previous ejaculate" (to avoid confusion with delay from the insemination with the prior male's ejaculate)

L 145: The variable about the delay between ejaculates may also partly reflect sperm depletion effects, not only sperm aging.

L 190-193: it sounds as if there is some sort of control in the machine learning algorithm for repeated observations of the same male. It would also be appropriate to include some sort of grouping variable accounting for multiple eggs in the same clutch, and for repeated use of the same female.

L279: please use wording more similar to the earlier descriptions of this variable, to make it clear that it's the breeding attempts in his pedigree rather than that he himself has performed

L295: soften this statement (e.g., add "within the range of values explored")

L302: although (Vuarin et al., 2018) find no effect of inbreeding...

L330-333: the delay between the two ejaculates could also affect the relative number of stored sperm from the last vs. earlier inseminations (as indicated in L311-313)—so this isn't necessarily strictly an aging effect

L 388: "survive" or "succeed" would sound more appropriate than "strive" here

Figure 2, 3: please indicate in the legend that the insemination order was inversed (as indicated in Table 1 and L 249-250)

L637, 643: I don't understand how each point could correspond to an individual male, rather than an individual insemination? (based on a clutch size of 1-3 eggs, and the number of individuals and eggs mentioned, there must be multiple inseminations per male, and then these different inseminations would have different x-axis values?)

Please provide information about temperature control during measuring motility of the ejaculate (supplement)

Please explain a bit more how the number of breeding events in captivity differs from the number of generations in captivity (supplement)

Literature cited

Birkhead, T. R. (1991). Sperm depletion in the bengalese finch, *Lonchura striata*. *Behavioral Ecology*, 2, 267–275.

Johnsen, A., Carter, K. L., Delhey, K., Lifjeld, J. T., Robertson, R. J., & Kempenaers, B. (2012). Laying-order effects on sperm numbers and on paternity: comparing three passerine birds with different life histories. *Behavioral Ecology and Sociobiology*, 66, 181–190.

Kvarnemo, C., & Simmons, L. W. (2013). Polyandry as a mediator of sexual selection before and after mating. *Philosophical Transactions of the Royal Society of London. Series B, Biological Sciences*, 368, 20120042.

Vuarin, P., Bouchard, A., Lesobre, L., Levêque, G., Chalah, T., Saint Jalme, M., ... Sorci, G. (2018). No evidence for prezygotic postcopulatory avoidance of kin despite high inbreeding depression. *Molecular Ecology*, 27, 5252–5262.

Reviewer #3 (Remarks to the Author):

This is an excellent study which uses a wealth of data and long-term expertise of a unique avian model system to carefully flesh out the relative importance of multiple parameters in the outcome of sperm competition. The results are very interesting and make a substantive contribution to the field. The ms will attract a broad readership ranging from those interested in sexual selection and sexual conflict to those interested in animal production and reproductive physiology.

As a general comment, previous work (e.g. by Tim Birkhead) indicated that the time of day of an insemination can play an important role in sperm precedence in artificial insemination studies, largely due to the fact that inseminations around the time of egg-laying are at a disadvantage. It would be interesting to report the time of day that birds were inseminated in this study, and how this relates to the typical time of egg-laying in this species. If there is considerable variation in insemination time, it may be worth testing whether adding this variable increases the explanatory power of the model.

The findings of last male precedence are broadly consistent with those reported by Carleial et al (2020) (ref [21] in the ms) for natural copulations in red junglefowl. It may be worth mentioning this in the discussion to lend support to the fact that the findings of this study may also be representative of patterns for natural copulations.

Congratulations on a great study!

Tom Pizzari

Reviewers' comments:

Reviewer #1 (Remarks to the Author):

In their paper 'Ranking the importance of factors driving siring success during sperm competition', Sorci and colleagues used a large dataset from a long-term breeding program in houbara bustards for a comprehensive assessment of the factors potentially contributing to competitive fertilization success. They included a wide range of traits reflecting ejaculate quality, ejaculation and insemination history, and male age or sexual display and used a powerful statistical approach to estimate the relative contributions of these different parameters to the paternity of each egg. Male insemination order and female sperm storage duration before fertilization had the strongest effect on paternity, followed by sperm motility and number.

This study makes an important contribution to the field by showing the multifarious nature of competitive fertilization success without the rigorous experimental control of confounding variables (but possibility of statistical control), which potentially captures a more natural scenario than in previous studies.

Response: Thank you for these positive comments.

Some points I would like the authors to consider are:

1. The authors mentioned male-female interactions (or female biases among sperm from different males) in their introduction, which is an important issue but not addressed in the analyses.

a. Is there any information on female age, fecundity, reproductive history or other that could be examined along with the male traits?

Response: Thank you for this suggestion. We have included female age in the model. We did not include reproductive history or fecundity because they are strongly correlated with female age (all females essentially breed every year and the overall number of eggs laid obviously increases from one year to the other). That said, we would like to recall that including female age in the model can only have effects on the other predictors, since, by no way, female age could directly affect the response variable. The reason is that, while for a given egg male predictors can take different values, any female trait would be an invariant. In agreement with this, the results of the new model confirmed that the "importance" of female age as a predictor of fertilization success was nil. In addition, we did not find strong evidence suggesting that including female age changed the ranking (and the weight) of the other predictors.

b. Do some females consistently show a stronger bias between males? Are the replicated combinations between males and females that could be compared (e.g., repeatability within combinations, consistent ranking between combinations)?

Response: This is a very interesting point. We would have liked to test this. Nevertheless, our breeding strategy consists in optimizing the genetic diversity and maximizing the number of artificial inseminations performed each day. In addition, considering the high number of birds and inseminations in the breeding project, it is highly improbable to have the same combinations of males repeated several times. Indeed, only two females were inseminated with the same combinations of males precluding any possible analysis of the repeatability of fertilization success within combinations.

2. In many species, males have been shown to plastically adjust sperm number or quality to their social context, be it presence of other males or female quality.

a. How many different dummy females were used and how do they compare?

Response: Many dummies have been used across years. They are extremely simple as shown in the picture here below.

b. Is there any effect of dummy ID on ejaculate traits that could indicate strategic ejaculate allocation and thus variation between samples?

Response: Given their “simplicity”, the dummies do not have a specific ID and therefore we cannot explore whether males consistently allocate more or less sperm across different dummies. But again, given the extreme simplicity of the dummies, it seems reasonable to assume that males would not have much cues to adopt different allocation strategies.

c. If not, do you have any data on natural matings that would allow you to infer the importance of male-female combinations?

Response: It is very difficult to have data on natural matings beyond anecdotic observations of the display males perform and the behavior females express when mating. Unfortunately, no data are available to properly address the question of male-female compatibility in the wild for this species.

d. Finally, are males exposed to other males before or during semen collection that could affect the samples?

Response: Males are housed in individual pens and do not have direct contact with other individuals that might affect how males “mate” with the dummy (i.e., no direct disturbance). However, although there are no physical interactions among males, they are visually and acoustically exposed to other males. Given that all males in the breeding unit are possibly in interaction (visually and acoustically), it was not possible to include this parameter in the present analysis.

3. The authors repeatedly refer to 13 or 14 ‘phenotypic’ traits that they measured. However, I would not consider variables like mating order, Julian date or (artificial) insemination quality ‘phenotypic traits’. How about ‘variables’ or ‘parameters’, or else counting the phenotypic traits and those simply used as control variables?

Response: We agree with the referee that mating order cannot be considered as a phenotypic trait. We used the term “parameter” to refer to the predictors included in the model.

Some more specific comments (by line):

L26 ‘of’ instead of ‘from’ after independent

Response: Corrected.

L40 It would be fair to acknowledge Parker (1970) here, too

Response: Done.

L43 ‘contest’ sounds a bit odd for sperm competition. The also, due to the generality of this phenomenon, you could also consider citing some broad reviews (e.g., Pizzari and Parker 2009;

Parker and Pizzari 2010) instead of a random mix of intraspecific and comparative case studies (which themselves are not necessarily directly comparable).

Response: We agree that the word contest is not the most appropriate one. Accordingly, the sentence now reads "... to outcompete the rivals". We also cited the suggested broad reviews instead of the selected primary intraspecific and comparative studies.

L48-50 Note that a follow-up paper by the same team even specifically addressed those complex interactions between competing sperm and the female reproductive tract (Lüpold et al. 2020).

Response: Thank you for this suggestion, we have added this reference in the revised version.

L50 I think 'hostile' should be avoided as it is a very loaded term. How about 'selective', 'challenging' or anything along those lines? If you choose the former, it also makes it immediately clear why some males' sperm might be favored over others as mentioned in the rest of this sentence.

Response: We agree and, as suggested, we replaced the word "hostile" with the term "selective".

L138 Better: "For each egg included..."

Response: Corrected.

L187 'technique' needed'? If yes, plural? Does not quite work as it stands.

Response: Changed as: "BRT combines classification and regression tree models with boosting optimization techniques".

L201 'Hyperparameter tuning'

Response: Corrected.

L332 True, but more likely this enhanced advantage could result from the combination of sperm aging and sperm loss.

Response: Yes, we agree. This has been corrected accordingly.

L388 'released into the wild'

Response: Corrected.

L390 'start'

Response: Corrected.

L397 'in the long run'

Response: Corrected.

Literature cited:

Lüpold, S., J. B. Reil, M. K. Manier, V. Zeender, J. M. Belote, and S. Pitnick. 2020. How female × male and male × male interactions influence competitive fertilization in *Drosophila melanogaster*. *Evol. Lett.* 4:416–429.

Parker, G. A. 1970. Sperm competition and its evolutionary consequences in the insects. *Biol. Rev.* 45:526–567.

Parker, G. A., and T. Pizzari. 2010. Sperm competition and ejaculate economics. *Biol. Rev.* 85:897–934.

Pizzari, T., and G. A. Parker. 2009. Sperm competition and sperm phenotype. Pp. 207–245 in T. R. Birkhead, D. J. Hosken, and S. Pitnick, eds. *Sperm Biology: An Evolutionary Perspective*. Academic Press, San Diego.

Reviewer #2 (Remarks to the Author):

«Ranking the importance of factors driving siring success during sperm competition” uses an impressive dataset on artificially-inseminated houbara bustards, from a well-studied long-term captive breeding program, to examine how various factors relate to fertilization probability after competitive inseminations. The paper is well-written, and it provides a valuable contribution to understanding how different factors related to ejaculate timing and quality, and male quality, affect fertilization success in a bird. As the authors note, this type of data is quite rare for a long-lived vertebrate. They use a machine learning approach, which they explain clearly, to investigate this. I am not qualified to comment on the accuracy/rigor of the machine learning approach.

Response: Thank you for these positive comments.

1. I recommend that the authors present some follow-up traditional statistics and/or figures, perhaps as a supplement, to help ensure readers understand how to interpret the SHAP outputs (and also to be sure that readers are convinced by these outputs!) The authors justify the machine learning approach partly as a way to avoid difficult problems with multicollinearity, but it seems to me that one or two smaller, confirmatory model based on the machine learning results could be useful. For example, a model including either insemination order or insemination-laying delay (as they are highly correlated, Fig 2), as well as sperm motility and/or the sperm number in the ejaculate (which are more weakly correlated), and number of days the male displayed. Some of these variables might be converted to categorical variables based on the machine learning results (e.g., number of sperm in the ejaculate). Of course it is the variance inflation factor, rather than the raw correlation among these variables, that is important; perhaps including all 5 predictors in one model would be possible.

Response: We agree with the referee that a more traditional approach might help those who are not familiar with BRT. As suggested, we ran a generalized linear mixed model with the 5 predictors (the number of days with display was replaced by the percentage of days displaying, as recommended by the referee in one of the comments below). The model was run using R 3.6.0. (*glmmTMB* package) with a binomial distribution and the identities of the male and the egg as crossed intercept random effects. Multicollinearity was assessed using the *performance* package. We ranked the variables according to their parameter estimates (given that the predictors have been standardized, parameter estimates correspond to the effect sizes).

	Estimate	Standard Error	VIF
Male insemination order	-1.095	0.101	1.66
Delay between insemination and egg laying	-0.283	0.091	1.65
Mass motility index	0.169	0.057	1.12
Percentage of days displaying	-0.117	0.055	1.01
Number of sperm in the ejaculate	0.064	0.057	1.11

The ranking provided by the GLMM was actually very similar to the one given by the BRT, with insemination order and delay between insemination and egg laying having the largest effect sizes. The slight differences in ranking among the other variables with lower predictive power is not surprising given that the BRT model also includes the non-linear relationship between the predictors and the response variable, and the interactions between different predictors.

2. In a few places, the paper over-sells its ability to reflect selection on sperm in the wild (e.g., the contrast between lab and natural conditions at L 58-60; the fact that neither the abstract, the introduction, nor the opening paragraph of the discussion states that the study uses artificial insemination or places caveats on the findings related to that; L 334-339). While the authors do acknowledge that their study is a long way from wild conditions (L 341-349), several aspects of the study are over-interpreted with respect to their immediate relevance to wild populations, and this must be improved.

Response: We fully agree with the referee that these results cannot be directly extrapolated to infer how selection acts on ejaculate attributes and sperm traits in the wild. We wished to explicitly state this in the manuscript by saying that our study is a long way from wild conditions, and we are sorry that the referee had the perception that we over-sell the results, this was not our intent. To fully address the different points raised by the referee here, we added a section at the end of the discussion where the limitations of the study are clearly mentioned (such as those due to the captive conditions).

2A. For example, L 355-385 focuses on the proxy for pre-copulatory investment. In the wild, among-male variation in the ability to acquire resources could alter how (and if) investment into pre- and post-copulatory traits are correlated (see, e.g., Kvarnemo & Simmons, 2013), while in the captive population, presumably there is only variation in allocation. It's therefore difficult to directly apply these results to wild animals. Also, L109-113: Please provide evidence that this measure reflects investment into pre-copulatory sexual selection, rather than, for example, variation in how well males tolerate captivity.

Response: Indeed, this is one of the examples where differences between captivity and natural conditions might affect the results. We explicitly stated this in the revised version. All birds used for this studies are born in captivity and human imprinted, which obviously reduces the potential stress due to captive conditions. That said, we cannot exclude that interindividual differences in personality (even among animals born in captivity) might account for differences in the propensity to display, but probably this also applies for birds displaying in the wild.

2B. Additionally, several variables relating to timing seem likely to be longer than in nature. Specifically, Table 1 indicates that the delay between insemination and egg laying is on average 11 days, which is quite a lot longer than in passerine birds (e.g., Johnsen et al., 2012). The number of days between ejaculations by the male is quite variable (up to 57 days, mean of about 4), and the delay between inseminations is on average 3.2 days, with a maximum of 23. Is there any information about these variables from wild populations of bustards, so that we know whether the values used are biologically relevant in a wild condition? Delay between successive inseminations only seems to have an effect when it has a value greater than about 8 days (Fig S1C), which is also about the point where inseminations become inherently less effective (delay between insemination and egg laying, Fig 3B), which suggests possible collinearity problems.

Response: As mentioned above, it is very difficult to gather accurate data on matings under natural conditions for this species. Therefore, we can only state again that the results should not be directly extrapolated to infer what happens in nature. That said, we think that referring to data on passerines is probably not a particularly relevant comparison for the houbara bustard. We know that females can store sperm in dedicated structures (sperm storage tubules) and data from the captive breeding indicate that females can lay fertile eggs up to 40 days post-insemination. But again, we do not know how frequently females (and males) mate in the wild. This point has been explicitly addressed in the "limitations" paragraph at the end of the discussion.

Concerning the possible collinearity between delay between successive inseminations and the delay between insemination and egg laying, we checked the VIFs and did not find strong evidence for collinearity (VIFs < 1.1).

2C. Finally, one additional element that should be mentioned is the possibility that female interactions with the male himself guide how females use his sperm (mentioned in general, e.g., L 66; see also citation 16 Løvlie et al.).

Response: We again agree with the referee that one of the caveats of our study is that artificial inseminations does not allow females to express any pre-copulatory choice, which might subsequently drive how female use the sperm of different males. We explicitly stated this in the "limitations" section, at the end of the discussion.

3. In addition, two variables should be altered and re-analyzed, and the authors should consider including additional measures of intrinsic ejaculate quality, such as morphological normality of the sperm (e.g. citation 11, Vuarin et al. 2019).

Response: Sperm morphology and viability are not assessed routinely following ejaculate collection, therefore including the data on these two variables would substantially reduce the sample size, precluding the use of BRT (which inherently needs large sample size to provide good predictive power).

3A. L 149, table 1: The number of ejaculates collected within the preceding year seems like much too broad a time span for examining sperm depletion. In Bengalese finches (admittedly a distant relative), recovery from sperm depletion occurs within 24 hours (Birkhead, 1991). I hope it is feasible to re-calculate this variable using a shorter time window; otherwise, please do not make conclusions about the role of sperm depletion from this variable (e.g., L 296).

Response: We are sorry if the definition of the variable that we used to infer sperm depletion was unclear. The variable refers to the total number of ejaculates collected prior to the one used to inseminate the female in the same year (not the preceding year). In other words, if the insemination occurred the first of March, the total number of ejaculates simply refers to the number of collections that had occurred between first January and first March.

3B. L 378-384, 109-113: Consider using the percent of days, rather than the number, when the male displays (from the beginning of the season), to avoid dependence between these two variables. If the number of days the male displays largely reflects the date when the sperm was sampled, it is difficult to interpret this as a proxy for investment into precopulatory traits.

Response: Thank you for this suggestion. We have now computed the percent of days with display (from the 1st of January) and used this variable in the model.

4. At L 413-414, we learn that the males were relatively young. Are there other ways in which the dataset is a biased subsample of males/inseminations? For example, in citation 59, the authors describe an experiment aimed at discovering age effects, which might result in the competing ejaculates being from more age-different males than would be expected by chance. Are there other similar factors at play?

Response: As any population, the captive flock has more young than old individuals. Therefore, we did not expect to have a uniform distribution of age among the individuals included in the database. However, we did not expect to see such a biased distribution towards young individuals ($90\% \leq 7$ yrs). We do not see any possible bias for the other parameters included in the model.

5. It strikes me as strange that the number of sperm in the ejaculate, but not the number of sperm inseminated, is a predictor of fertilization success (L259, 274). This suggests that it's not sperm number per se, in the sense of a sperm competition raffle, that is driving the pattern, but perhaps some other ejaculate quality trait that correlates with number of sperm ejaculated? This result is not highlighted in the discussion.

Response: Yes, we were also surprised. We agree with the referee that one possible explanation is that sperm number in the ejaculate correlates with another "quality" trait that we did not measure. Another interesting observation is that the shape of the relationship between fertilization success (SHAP values) and number of sperm in the ejaculate or number of sperm inseminated, looks pretty much the same (fertilization success increases and then reaches a plateau where adding sperm does not further improve fertilization success). As suggested, we have discussed this result in the revised version.

Minor comments

L 25, 163: not all are phenotypic traits (e.g., mating order, date); perhaps "factors" would work instead?

Response: Yes, we agree that phenotypic trait is not appropriate for all predictors. We used the term “parameter” instead.

L 26: Similarly, I was confused at first reading about what you meant by independent from male quality (particularly since mating order might correlate with male quality in wild populations, and the abstract does not state that this was a study using artificial insemination). It might read better if you instead re-word and describe all 3 non-intrinsic factors that were important.

General methods section (or perhaps in the supplement): please describe briefly the housing conditions of the birds. Are males isolated from females? What kind of enclosure are they in?

Response: As suggested we stated in the abstract that the study is based on artificial insemination; any variation in “mating” order cannot reflect male quality simply because females do not choose with whom to mate. We also described the housing conditions in the supplement. Birds are housed individually in outdoor 2 x 2m pens with gravel substrate. Therefore, they are not in direct contact other individuals (except from visual and acoustic contact).

L 90-92: Clarify that the ECWP is in Morocco, to clarify why the animal care approval from Morocco is the relevant approval

Response: Done.

L99-100, 132-134: Can you please provide information about how many males' sperm was inseminated for each egg, and perhaps give a range of what percent of “attempted” eggs each individual male succeeded in fertilizing?

Response: The average number of inseminated sperm was 19×10^6 (SD = 10×10^6). The average fertilization success (fertilized/attempted) was 38.7% (SD = 40.12%). This has been added in the revised version.

L105: “on the day of laying, all eggs” implies that the female lays multiple eggs in one day. Is that correct for this species? If not, perhaps reword to “Each egg is collected on the day it is laid...”

Response: Sorry for the awkward writing. This should indeed read “Each egg is collected on the day it is laid”.

L 134 (or elsewhere): please clarify whether predictor variables (e.g. sperm motility index) were evaluated separately for each insemination by a particular male (not using a previous value from that male)

Response: Indeed, values of predictor variables (such as the motility index) refer to each insemination. In other words, values never refer to previous ejaculate assessments. This has been added in the revised version.

L 142-162, Table 1: The authors have done a really good job of keeping organized with so many different predictor variables. This could be further improved if they are presented in the same order in the text and in the table. “delay with previous ejaculate” might be more clear as “Delay since male’s previous ejaculate” (to avoid confusion with delay from the insemination with the prior male’s ejaculate)

Response: Thank you for this suggestion. We have corrected the text accordingly.

L 145: The variable about the delay between ejaculates may also partly reflect sperm depletion effects, not only sperm aging.

Response: Yes, indeed. We have corrected this in the revised version.

L 190-193: it sounds as if there is some sort of control in the machine learning algorithm for repeated observations of the same male. It would also be appropriate to include some sort of grouping variable accounting for multiple eggs in the same clutch, and for repeated use of the same female.

Response: Actually, we did not include the repeated use of the same female because this does not affect the outcome of the model (by definition all females laid fertilized eggs here because this was one of the inclusion criteria used to build the dataset). To confirm this, we ran a GLMM where, in addition to male ID, we also included the egg ID nested within the female ID. This model had higher AIC and BIC values compared to the simpler model with male and egg ID as crossed intercept random effects. We therefore decided to use this simpler model structure for the BRT.

	Random effects	df	AIC	BIC
Model 1	1 Egg_ID + 1 Male_ID	9	2496.9	2542.6
Model 2	1 Female/Egg_ID + 1 Male_ID	8	2498.9	2550.3

L279: please use wording more similar to the earlier descriptions of this variable, to make it clear that it's the breeding attempts in his pedigree rather than that he himself has performed

Response: Corrected.

L295: soften this statement (e.g., add “within the range of values explored”)

Response: Added.

L302: although (Vuarin et al., 2018) find no effect of inbreeding...

Response: Yes, indeed, but beyond relatedness, compatibility might involve specific loci (e.g., MHC).

L330-333: the delay between the two ejaculates could also affect the relative number of stored sperm from the last vs. earlier inseminations (as indicated in L311-313)—so this isn't necessarily strictly an aging effect

Response: We agree, this also reflects a sort of passive sperm loss. We added this in the revised version.

L 388: “survive” or “succeed” would sound more appropriate than “strive” here

Response: Changed.

Figure 2, 3: please indicate in the legend that the insemination order was inversed (as indicated in Table 1 and L 249-250)

Response: Done.

L637, 643: I don't understand how each point could correspond to an individual male, rather than an individual insemination? (based on a clutch size of 1-3 eggs, and the number of individuals and eggs mentioned, there must be multiple inseminations per male, and then these different inseminations would have different x-axis values?)

Response: Sorry for the mistake, this is indeed individual insemination. We corrected it in the revised version.

Please provide information about temperature control during measuring motility of the ejaculate (supplement)

Response: Motility is assessed routinely at room temperature, soon after the ejaculate is collected and before the insemination is done. Keeping the ejaculate at room temperature during the few minutes that precede the insemination does not affect the insemination success.

Please explain a bit more how the number of breeding events in captivity differs from the number of generations in captivity (supplement)

Response: We added the following diagram as to better explain the difference between the number of breeding events and the number of generations in captivity

If we assume that individuals A to H are the six founders of a hypothetical captive flock (generation 0), the individual O has generation 3, but it is born after 7 breeding events.

Literature cited

Birkhead, T. R. (1991). Sperm depletion in the bengalese finch, *Lonchura striata*. *Behavioral Ecology*, 2, 267–275.

Johnsen, A., Carter, K. L., Delhey, K., Lifjeld, J. T., Robertson, R. J., & Kempenaers, B. (2012). Laying-order effects on sperm numbers and on paternity: comparing three passerine birds with different life histories. *Behavioral Ecology and Sociobiology*, 66, 181–190.

Kvarnemo, C., & Simmons, L. W. (2013). Polyandry as a mediator of sexual selection before and after mating. *Philosophical Transactions of the Royal Society of London. Series B, Biological Sciences*, 368, 20120042.

Vuarin, P., Bouchard, A., Lesobre, L., Levêque, G., Chalah, T., Saint Jalme, M., ... Sorci, G. (2018). No evidence for prezygotic postcopulatory avoidance of kin despite high inbreeding depression. *Molecular Ecology*, 27, 5252–5262.

Reviewer #3 (Remarks to the Author):

This is an excellent study which uses a wealth of data and long-term expertise of a unique avian model system to carefully flesh out the relative importance of multiple parameters in the outcome of sperm competition. The results are very interesting and make a substantive contribution to the field. The ms will attract a broad readership ranging from those interested in sexual selection and sexual conflict to those interested in animal production and reproductive physiology.

Response: Thank you very much for your positive comments.

As a general comment, previous work (e.g. by Tim Birkhead) indicated that the time of day of an insemination can play an important role in sperm precedence in artificial insemination studies, largely due to the fact that inseminations around the time of egg-laying are at a disadvantage. It would be interesting to report the time of day that birds were inseminated in this study, and how this relates to the typical time of egg-laying in this species. If there is considerable variation in insemination time, it may be worth testing whether adding this variable increases the explanatory power of the model.

Response: Thank you for this suggestion. All inseminations are done during the morning (between 8 and 11 am).

The findings of last male precedence are broadly consistent with those reported by Carleial at al (2020) (ref [21] in the ms) for natural copulations in red junglefowl. It may be worth mentioning this in the discussion to lend support to the fact that the findings of the this study may also be representative of patterns for natural copulations.

Response: Thank you for this suggestion. We included it in the discussion of the revised version.

Congratulations on a great study!
Tom Pizzari

Reviewers' comments:

Reviewer #1 (Remarks to the Author):

The authors have greatly improved their manuscript, but a few minor issues would be worth addressing:

General comment on format: I am aware that first submission does not need to adhere to the journal style guide. But currently the methods are described before the results. Since the methods would come last in Nature-style journals (to my knowledge also in Communications Biology), the results would currently not be interpretable without first going to the end of the paper for even the most basic methods information on the data. It is typical that such papers provide at least minimal context to the results themselves, with more detail in the methods. I leave this at the discretion of the editor and authors, but if this paper really needs a different format, it might require another round of reviews to ensure coherence. If so, it could have helped to change all this before resubmission as it would have to happen anyway.

Title: I would recommend adding the study species to the title or at the very least narrow it down to a bird. As it stands, it implies some sort of universal ranking, which it clearly isn't.

Line 24: It would help to add the scientific name to the common name, unless you do so in the title. It does not appear anywhere in the title, abstract or keywords, so the paper may be harder to find by searches based on the species.

Line 42: "exerts selection" seems sufficient, and it also states more clearly that it is a selective process itself rather than only putting on pressure.

Line 234: insert "the" before "SHapley...technique", and then again before "SHAP approach" on line 237

Line 294 (and elsewhere): "Julian date" is not the days since January 1st but the days since the beginning of the Julian Period

Line 335: "consecutive" might be better here than "subsequent" because the latter refers to the inseminations following the focal one. This is not an issue in most context, but here it is confusing immediately after "the last mating male" (i.e. there should have been no further mating thereafter, and you are referring to the interval before the last mating).

Line 364: This might be subtle, you maybe consider "acknowledge" or "are aware" instead of "agree"? Just because agreeing tends to imply that you mentioned this caveat in response to a reader's (or reviewer's) questioning it. But from what I can tell (at least with this journal), you seem to have taken your own initiative in addressing it and so could take credit for your critical assessment.

Line 394: I would say "were able to include" instead of "could include" as the latter sounds too much like a possibility ("we could if we wanted or had to...").

Reviewer #2 (Remarks to the Author):

The authors have thoroughly and satisfactorily addressed my previous concerns. I have a few minor suggestions.

L 166: insert Julian before day of insemination, for consistency with language used in results (L 294)

L 184 and 677: please indicate what type of correlation (Spearman, Pearson?), and for L 677, indicate that only squares below the diagonal contain data (I presume)

L 329: please make it clear whether the Houbara bustard is such a species (I would have assumed that they lack an intromittent organ, also implied at L 464?)

Reviewer #1 (Remarks to the Author):

The authors have greatly improved their manuscript, but a few minor issues would be worth addressing:

Response: Thank you for your positive comments.

General comment on format: I am aware that first submission does not need to adhere to the journal style guide. But currently the methods are described before the results. Since the methods would come last in Nature-style journals (to my knowledge also in Communications Biology), the results would currently not be interpretable without first going to the end of the paper for even the most basic methods information on the data. It is typical that such papers provide at least minimal context to the results themselves, with more detail in the methods. I leave this at the discretion of the editor and authors, but if this paper really needs a different format, it might require another round of reviews to ensure coherence. If so, it could have helped to change all this before resubmission as it would have to happen anyway.

Response: We moved the method section as to adhere to the journal style guide.

Title: I would recommend adding the study species to the title or at the very least narrow it down to a bird. As it stands, it implies some sort of universal ranking, which it clearly isn't.

Response: We added the study species in the title. We note, however, that the title has now 17 words.

Line 24: It would help to add the scientific name to the common name, unless you do so in the title. It does not appear anywhere in the title, abstract or keywords, so the paper may be harder to find by searches based on the species.

Response: We added the scientific name, as suggested.

Line 42: "exerts selection" seems sufficient, and it also states more clearly that it is a selective process itself rather than only putting on pressure.

Response: Corrected.

Line 234: insert "the" before "SHapley...technique", and then again before "SHAP approach" on line 237

Response: Corrected.

Line 294 (and elsewhere): "Julian date" is not the days since January 1st but the days since the beginning of the Julian Period.

Response: We deleted "Julian".

Line 335: "consecutive" might be better here than "subsequent" because the latter refers to the inseminations following the focal one. This is not an issue in most context, but here it is confusing immediately after "the last mating male" (i.e. there should have been no further mating thereafter, and you are referring to the interval before the last mating).

Response: Corrected.

Line 364: This might be subtle, you maybe consider “acknowledge” or “are aware” instead of “agree”? Just because agreeing tends to imply that you mentioned this caveat in response to a reader’s (or reviewer’s) questioning it. But from what I can tell (at least with this journal), you seem to have taken your own initiative in addressing it and so could take credit for your critical assessment.

Response: We replaced “agree” with “acknowledge”.

Line 394: I would say “were able to include” instead of “could include” as the latter sounds too much like a possibility (“we could if we wanted or had to…”).

Response: Corrected.

Reviewer #2 (Remarks to the Author):

The authors have thoroughly and satisfactorily addressed my previous concerns. I have a few minor suggestions.

Response: Thank you for your positive comments.

L 166: insert Julian before day of insemination, for consistency with language used in results (L 294)

Response: Following the suggestion of referee 1, we removed “Julian” from the text.

L 184 and 677: please indicate what type of correlation (Spearman, Pearson?), and for L 677, indicate that only squares below the diagonal contain data (I presume)

Response: We indicated that the correlation matrix reports Pearson correlation coefficients, and that the coefficients are reported below the diagonal in the figure.

L 329: please make it clear whether the Houbara bustard is such a species (I would have assumed that they lack an intromittent organ, also implied at L 464?)

Response: The houbara bustard lacks an intromittent organ. We explicitly stated it.